# Long-Term Prognostic Value of Automated Measurements in Nuclear Cardiology: Comparisons with Expert Scoring

**DOI:** 10.3390/medicina59101738

**Published:** 2023-09-28

**Authors:** George Angelidis, Stavroula Giannakou, Varvara Valotassiou, Ioannis Tsougos, Chara Tzavara, Dimitrios Psimadas, Evdoxia Theodorou, Anastasia Ziaka, Charalampos Ziangas, John Skoularigis, Filippos Triposkiadis, Panagiotis Georgoulias

**Affiliations:** 1Nuclear Medicine Laboratory, University Hospital of Larissa, University of Thessaly, Mezourlo, 41110 Larissa, Greece; 2Medical Physics Laboratory, University Hospital of Larissa, University of Thessaly, Mezourlo, 41110 Larissa, Greece; 3Department of Cardiology, University Hospital of Larissa, University of Thessaly, Mezourlo, 41110 Larissa, Greece

**Keywords:** automated analysis, myocardial perfusion imaging, prognosis, summed difference score, summed rest score, summed stress score

## Abstract

*Background and Objectives*: Automated methods for the analysis of myocardial perfusion studies have been incorporated into clinical practice, but they are currently used as adjuncts to the visual interpretation. We aimed to investigate the role of automated measurements of summed stress score (SSS), summed rest score (SRS), and summed difference score (SDS) as long-term prognostic markers of morbidity and mortality, in comparison to the prognostic value of expert reading. *Materials and Methods*: The study was conducted at the Nuclear Medicine Laboratory of the University of Thessaly, in Larissa, Greece. A total of 378 consecutive patients with known or suspected coronary artery disease were enrolled in the study. All participants were referred to our laboratory for the performance of stress/rest myocardial perfusion single photon emission computed tomography. Automated measurements of SSS, SRS, and SDS were obtained by Emory Cardiac Toolbox (ECTb (Version 3.0), Emory University, Atlanta, GA, USA), Myovation (MYO, Xeleris version 3.05, GE Healthcare, Chicago, IL, USA), and Quantitative Perfusion SPECT (QPS (Version 4.0), Cedars-Sinai Medical Center, Los Angeles, CA, USA) software packages. Follow-up data were recorded after phone contacts, as well as through review of hospital records. *Results*: Expert scoring of SSS and SDS had significantly greater prognostic ability in comparison to all software packages (*p* < 0.001 for all comparisons). Similarly, ECTb-obtained SRS measurements had significantly lower prognostic ability in comparison to expert scoring (*p* < 0.001), while expert scoring of SRS showed significantly higher prognostic ability compared to MYO (*p* = 0.018) and QPS (*p* < 0.001). *Conclusions*: Despite the useful contribution of automated analyses in the interpretation of myocardial perfusion studies, expert reading should continue to have a crucial role, not only in clinical decision making, but also in the assessment of prognosis.

## 1. Introduction

Myocardial perfusion imaging (MPI) has a crucial role in the management of patients with known or suspected coronary artery disease (CAD). Automated methods for the analysis of medical data have gained significant attention in the field of imaging modalities, including nuclear cardiology applications. The major advantage of computer-based automated analysis is the ability to exclude the subjective character of interpretation when evaluating MPI studies. Notably, the traditional visual assessment of MPI studies requires extensive reading experience. Apart from the relative tracer uptake on images, several additional factors should be taken into account, such as the pre-test likelihood of CAD, image quality, and potential attenuation artefacts. Therefore, a prolonged period of training is required for the comprehensive assessment of these factors, and the interpretation of MPI studies continues to be observer-dependent.

On the other hand, a number of software packages are commercially available in order to assist clinical decision making in nuclear cardiology. These packages can permit the analysis of MPI images based on normal databases [1,2,3]. According to the 17-segment model of the left ventricle, regional perfusion scores are derived using the average defect severity in a given segment [4]. Particularly, severity scores are automatically assigned to segments based on a five-point scale [5]. Subsequently, segmental scores can be summed either per region, or for the whole myocardium (summed stress score (SSS), summed rest score (SRS), summed difference score (SDS)). However, computer-based MPI analysis is currently used only as an adjunct to visual evaluation, and there are only scarce data regarding the incremental clinical value of automated measurements in nuclear cardiology [6,7,8]. Previously, we investigated the correlation between automated values and reader scoring of SSS, SRS, and SDS, as well as their associations with angiographic score, using coronary angiography as the gold standard [9]. 

In the present study, using three widely available software packages (Emory Cardiac Toolbox (ECTb (Version 3.0), Emory University, Atlanta, GA, USA), Myovation (MYO, Xeleris version 3.05, GE Healthcare, Chicago, IL, USA), Quantitative Perfusion SPECT (QPS (Version 4.0), Cedars-Sinai Medical Center, Los Angeles, CA, USA)), we aim to evaluate the clinical value of automated measurements of SSS, SRS, and SDS as long-term prognostic markers of morbidity and mortality in patients with known or suspected CAD, in comparison to the prognostic value of expert reading. 

## 2. Materials and Methods

### 2.1. Study Population

The present study was conducted at the Nuclear Medicine Laboratory, University Hospital of Larissa, Greece, between January 2014 and December 2018. A total of 378 consecutive patients, with known or suspected CAD, were enrolled in the study, as they did not meet any of the exclusion criteria and their follow-up data was available. All participants were referred to our laboratory for the performance of stress/rest myocardial perfusion single photon emission computed tomography (SPECT) and underwent coronary angiography prior to or after SPECT MPI (within a 3-month period). Patients gave informed consent for their complete enrolment, in compliance with the Hospital Ethics Committee guidelines and the ethical guidelines of the Declaration of Helsinki. Furthermore, written information regarding radiation protection was given to each participant. 

Medical history (mainly, clinical features, medications, previous cardiac events, CAD risk factors, and cardiac or non-cardiac comorbidities) was taken based on a brief structured interview. The presence of diabetes mellitus or lipid disorders was investigated in participants according to their medical history, including the use of the corresponding medications. Hypertension was defined as a systolic blood pressure of 140 mmHg or greater at rest and/or a diastolic blood pressure of 90 mmHg or greater at rest, or treatment with antihypertensive agents. Further, obesity was considered as a condition with body mass index (BMI) of 30.0 or greater (BMI calculated as weight in kilograms divided by height in metres squared).

Patients without proper withdrawal of cardio-active medications (i.e., b-blockers, calcium channel antagonists, and/or nitrates for approximately five half-lives) were excluded from the study, as these agents can influence performance during stress testing, MPI, and the associated parameters [5,10]. Moreover, we did not enrol patients with percutaneous coronary intervention (PCI) or coronary artery bypass grafting (CABG) ≤ 3 months after MPI, as cases of early revascularization are closely linked to MPI findings and should not be taken into account in the prognostic analysis. Moreover, we excluded patients with a history or other evidence of myocardial infarction. These patients comprise a heterogeneous group whose MPI data are affected not only by the presence of myocardial ischemia, but also by necrosis linked to both episode severity and applied therapy. Other exclusion criteria were severe congenital or valvular heart disease, non-ischemic cardiomyopathy, as well as pregnancy. Finally, we excluded patients with qualitatively suboptimal scintigrams, due to artefacts. 

### 2.2. Stress Testing

Symptom-limited treadmill testing (according to the Bruce protocol) was performed in 202 patients after 6 h- to 12 h-fasting and avoidance of smoking or heavy intense physical activity for at least 3 h. Data on symptoms related to the performance of exercise testing, and estimated workload in metabolic equivalents (METs, using standard tables) were recorded. In addition, 171 patients with contraindication or inability to achieve a satisfactory exercise level underwent pharmacologic testing (using adenosine or regadenoson), combined with low-level exercise. Pharmacologic stress without any form of exercise was performed in five patients with left bundle branch block (LBBB) or an implantable pacemaker. All stress testing procedure were in accordance with the European Association of Nuclear Medicine (EANM) guidelines [10,11]. 

### 2.3. Coronary Angiography and Angiographic Score

All coronary angiographies had been requested by cardiologists based on the medical history and clinical features of the patients. The studies were blindly interpreted by one experienced observer. Hemodynamically significant stenosis was defined as a stenosis of the vessel lumen greater than 50% (or fractional flow reserve ≤0.8), while the presence of a stenosis in the left mainstem was considered equivalent to a two-vessel disease. Consequently, the scoring of the angiographic studies (angiographic score) was made according to the following 4-point system: 0: normal study, 1: one-vessel disease, 2: two-vessel disease, 3: three-vessel disease. 

### 2.4. SPECT MPI & Semi-Quantification

MPI studies were carried out in the supine position, using a dual-headed SPECT camera, without attenuation-scatter correction. Technetium 99 m (99 mTc) tetrofosmin (Myoview, GE Healthcare, Chicago, IL, USA) was administered to all participants; injected activities were 250–400 MBq for stress and 625–1000 MBq for rest acquisitions according to societal guidelines [10]. Polar and three-dimensional mapping were performed (GE Xeleris Software (version 3.05), Milwaukee, WI, USA), and filtered back projection with the Butterworth Filter was used for tomographic reconstruction. 

Acquired and reconstructed data of both stress and rest imaging were blindly evaluated by two independent experienced observers. Radiotracer uptake was scored in each of the 17 LV segments using a 5-point scoring system (0: normal uptake; 1: mildly decreased uptake; 2: moderately decreased uptake; 3: severely decreased uptake; and 4: no uptake) [4]. Subsequently, the values of SSS and SRS were calculated by adding the scores of each segment in stress and rest imaging, respectively, while SDS was obtained by subtracting SRS from SSS [4]. In regions with decreased counts attributed to attenuation artefacts, the score was 0 [12]. Further, due to discordance between the two observers in 18 studies, the view of a third observer was requested and the disagreement was resolved by consensus [13]. 

In addition, for ECTb and QPS software packages, we created an institutional normal database from 100 patients (50 males and 50 females) with a low likelihood of CAD and visually normal stress and rest images, who were referred to our laboratory for the performance of SPECT MPI (not participated in the study). On the other hand, MYO software package does not have a user normal database creation feature. The automated measurements of SSS, SRS, and SDS were recorded for the participants, as derived by ECTb and QPS software packages (using the institutional normal database for each programme). Notably, MYO software package does not provide standardised segmental perfusion scores, and we converted the average segmental count values (relative to maximum pixel values in the relevant polar plot) to categorical scores according to >70%, 50–69%, 30–49%, 10–29%, and <10% thresholds, as previously described [14]. 

### 2.5. Follow-Up

Follow-up data were recorded after phone contact to the participants in the study, their relatives, or patients’ general practitioner or cardiologist, as well as through review of the patients’ hospital records. All-cause death, cardiovascular death, and non-fatal myocardial infarction were considered as hard events, while revascularization (≥3 months after myocardial SPECT, either PCI or CABG), hospitalizations (due to unstable angina, heart failure, or resuscitated cardiac arrest), and stroke as soft events. Cardiovascular death was defined according to the Tenth International Classification of Diseases (numbers I00–I99) as the death caused by diseases of circulatory system. In the presence of multiple causes, a reviewer, who was blinded to the hypothesis of the study, the clinical parameters, and the imaging data of the patients, evaluated the cause of death by examining the death certificates. All participants in the study were followed up for at least 36 months. 

### 2.6. Statistical Analysis

Quantitative variables were expressed as mean (standard deviation) or as median (interquartile range). Qualitative variables were expressed as absolute and relative frequencies. ROC curves (receiver operating characteristic curves) were used in order to estimate the prognostic ability of SSS, SRS, and SDS indexes for any cardiac event. Sensitivity and specificity were calculated for optimal cut-offs. The area under the curve (AUC) was also calculated. Kaplan–Meier survival estimates for any cardiac event were graphed over the follow-up period. Multivariate Cox proportional hazard model was used in order to determine the association of SSS, SRS, and SDS indexes with any cardiac event, after adjusting for age, gender, number of risk factors, and comorbidity. The assumption of proportional hazards was evaluated by testing for interaction with a continuous time variable. Hazard ratios (HR) with 95% confidence intervals (95% CI) were computed from the Cox regression analyses. All reported *p* values are two-tailed. Statistical significance was set at *p* < 0.05, and analyses were conducted using SPSS statistical software (version 26.0).

## 3. Results

The study population consisted by 378 patients (61.9% males) with a mean age of 63.8 years (SD = 9.6 years). Their characteristics are presented in Table 1. Soft cardiovascular events occurred in 31.7% of the sample, hard events in 11.6%, while any event occurred in 36.5%. Mean time until any event was 67.7 months (SE = 2.69 months). Patients’ survival curve according to Kaplan–Meier method is presented in Figure 1. 

Descriptives of SSS, SRS, and SDS are presented in Table 2, for each method separately (ECTb, MYO, and QPS software packages, and expert scoring). ROC analysis revealed that SSS and SDS had significant prognostic ability regarding “any event’’ based on all methods, while SRS had significant prognostic ability for “any event’’ in MYO and expert scoring (Table 3). Optimal cut-offs for SSS ranged from 4.5 (in expert scoring) to 11.5 (in ECTb). Optimal cut-offs for SDS ranged from 2.5 (in QPS) to 5.5 (in ECTb). As far as, SRS optimal cut-off was 4.5 in MYO and 1.5 in expert scoring. 

Based on the analysis of ECTb automated measurements, we found that SDS had significantly greater prognostic ability in comparison to SSS (*p* = 0.013) and SRS (*p* = 0.050), while SSS and SRS had similar prognostic ability (*p* > 0.05). As far as MYO software package, SSS had significantly greater prognostic ability compared to SDS (*p* < 0.001), while SRS had similar prognostic ability to SSS and SDS (*p* > 0.05). In QPS automated measurements, SSS and SDS had significantly greater prognostic ability compared to SRS (*p* < 0.001 and *p* = 0.011, respectively), while SSS and SDS had similar prognostic ability (*p* > 0.05). In expert scoring, SSS and SDS had significantly greater prognostic ability compared to SRS (*p* < 0.001 for both comparisons), while SSS and SDS had similar prognostic ability (*p* > 0.05). 

Subsequently, the prognostic ability of each index was compared between the four methods. Regarding SSS measurements, ECTb showed significantly lower prognostic ability in comparison to MYO (*p* < 0.001) and QPS (*p* = 0.005), while MYO and QPS had similar prognostic ability (*p* > 0.05). ECTb SRS measurements showed significantly lower prognostic ability in comparison to MYO (*p* < 0.001), whereas no significant difference was found compared to QPS (*p* > 0.05). Further, MYO SRS measurements had greater prognostic ability in comparison to QPS (*p* = 0.002). On the other hand, SDS prognostic ability was similar between all software packages (*p* > 0.05). 

Interestingly, regarding SSS values, ECTb, MYO, and QPS software packages had significantly lower prognostic ability compared to expert scoring (*p* < 0.001 for all comparisons). Moreover, ECTb SRS measurements showed significantly lower prognostic ability in comparison to expert scoring (*p* < 0.001), while expert scoring of SRS showed significantly greater prognostic ability compared to MYO (*p* = 0.018) and QPS (*p* < 0.001). Finally, regarding SDS values, expert scoring was linked to greater prognostic ability in comparison to all software packages (*p* < 0.001 for all comparisons) (Figure 2).

After conducting multiple Cox regression analyses, it was found that, after controlling for age, gender, risk factors, and comorbidity, greater SSS and greater SDS values (in all methods), as well as greater SRS values (in MYO and expert scoring), were significantly associated with greater hazard for “any event” (Table 4). 

## 4. Discussion

In the present study, with a long follow-up period of at least 36 months, we investigated the role of automated measurements of SSS, SRS, and SDS as long-term prognostic markers of morbidity and mortality, in comparison to the prognostic value of expert reading. An institutional normal database was created for the analysis using ECTb and QPS, while the vendor database was used for the analysis with MYO. Notably, expert scoring was demonstrated to have a significantly greater prognostic ability for all events under investigation compared to automated analyses. Moreover, based on our results, SSS and SDS had significant prognostic ability for “any event’’ (according to automated measurements and expert scoring), while SRS had significant prognostic ability for “any event’’ only in MYO software package and expert scoring.

Undoubtedly, MPI can offer valuable information in the prognostication of CAD patients. SPECT MPI has incremental prognostic value in patients with known or suspected CAD, and previous large studies and meta-analyses have demonstrated that a normal or low-risk myocardial perfusion SPECT is associated with a low annual cardiac event rate. Enrolling 4031 patients, Momose et al. revealed that SSS and SDS parameters were among the significant predictors of future events, in patients with known or suspected CAD [15]. Zhang et al. reported that SSS was the best independent predictor for hard cardiac events, whereas SDS was the strongest independent predictor for soft cardiac events [16]. Previously, in patients with chronic kidney disease, SDS was found to be a significant prognostic marker for major adverse cardiac, cerebrovascular, and renal events [17]. Also, higher SSS was linked to lower event-free survival rate [18]. Moreover, Bucerius et al. had reported that higher SSS is independently associated with cardiac events and lower incidence of cardiac event-free survival in the elderly [19]. However, in patients with severe left ventricular (LV) systolic dysfunction (LV ejection fraction ≤35%), Gimelli et al. showed that SSS is less predictive of outcome compared to patients with better preserved systolic dysfunction [20]. 

Interestingly, in a prospective study, Koh et al. investigated the long-term prognostic value of MPI classified as appropriate, according to the appropriate use criteria [21]. The researchers found that appropriate MPI was associated with long-term prognostic value for adverse events, despite a high proportion of low-risk patients (SSS ≤ 3). Further, using an ultrafast, low-dose MPI protocol in a cadmium–zinc telluride (CZT) camera, Lima et al. showed that CZT–MPI maintained the ability to stratify patients; larger areas of defect or ischemia were related to higher rates of hard events and late revascularization [22]. 

Currently, there are several software packages, such as ECTb, MYO, and QPS, for the automated analysis of myocardial perfusion in patients with known or suspected CAD. These algorithms aim to provide automated measurements of MPI parameters, in order to restrict the influence of reader’s experience over image interpretation. Nevertheless, the role of automated analysis continues to be adjunctive in clinical practice, mainly due to the inability of software packages to distinguish between real perfusion abnormalities and artefacts [9]. Furthermore, differences in the magnitudes of the automated measurements have been reported between software packages, despite the similar performance of the corresponding algorithms [23]. 

The cross-correlation of the outputs of different software packages has been investigated in several studies [23,24,25,26,27,28]. However, there are only few published data with regard to the correlation between automated quantitation and expert reading [9]. Previously, we have analysed the associations between automated measurements of SSS, SRS, and SDS with the expert reading, using coronary angiography as the gold standard [9]. Based on our results, visually defined SSS, SRS, and SDS were more strongly correlated to angiographic data compared to the software-derived corresponding values, supporting the significance of expert reading in MPI interpretation.

In the present study, we aimed to compare the prognostic performance between the automated analyses and expert reading of MPI studies, and we demonstrated the higher prognostic ability of expert scoring in comparison to all automated analyses. To the best of our knowledge, this is the first study aiming to compare the prognostic ability of automated analyses and expert scoring, with regard to myocardial perfusion parameters (SSS, SRS, and SDS). Based on our results, expert scoring is associated with significantly greater prognostic ability, supporting its value in patients’ prognostication.

## 5. Conclusions

MPI has an important role, not only in therapeutic management of patients with known or suspected CAD, but also for patients’ prognostication [29]. After comparing the prognostic ability between three widely available software packages and expert scoring of SSS, SRS, and SDS, we demonstrated the importance of visual assessment for patients’ prognostication. Our results indicated that current algorithms had a lower prognostic ability. However, possibly by implementing artificial intelligence methods, better algorithms may become available, providing advanced diagnostic investigation and prognostic assessment in CAD patients.

## Figures and Tables

**Figure 1 medicina-59-01738-f001:**
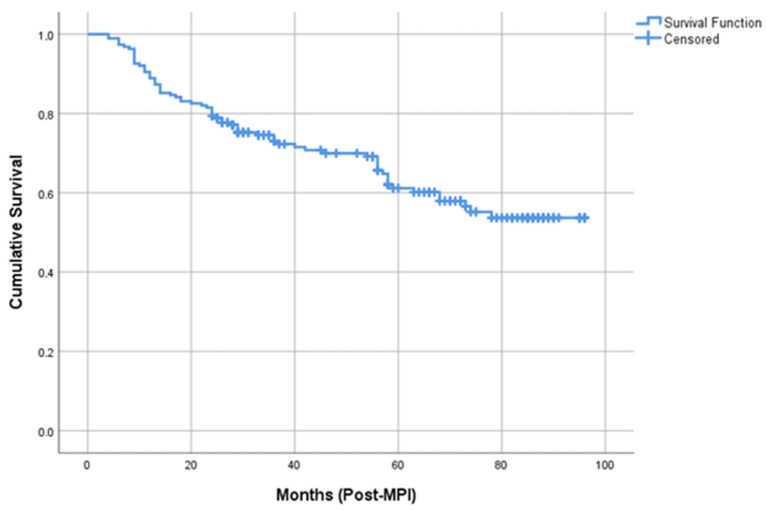
Kaplan–Meier curve for any cardiac event.

**Figure 2 medicina-59-01738-f002:**
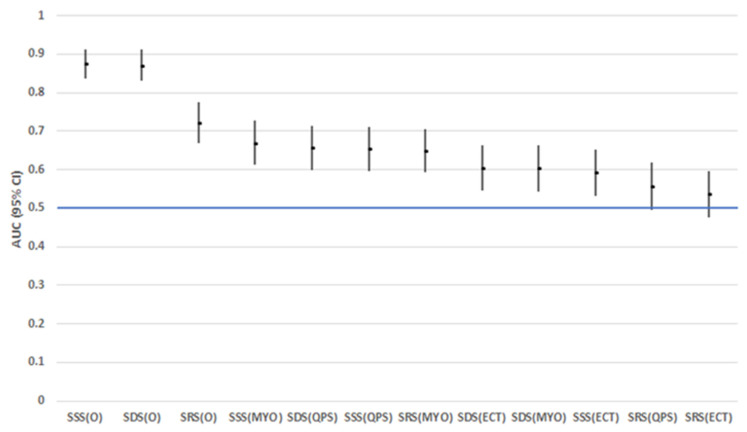
AUC (95% CI) for Summed Stress Score (SSS), Summed Rest Score (SRS), and Summed Difference Score (SDS), by descending order. ECT: Emory Cardiac Toolbox; MYO: Myovation; O: expert scoring; QPS: Quantitative Perfusion SPECT.

**Table 1 medicina-59-01738-t001:** Sample characteristics.

	N (%)
Gender	
Females	144 (38.1)
Males	234 (61.9)
Age, mean (SD)	63.8 (9.6)
BMI, mean (SD)	29.5 (5.4)
Symptoms	272 (72)
Angina	94 (24.9)
Angina-like symptoms	104 (27.5)
Dyspnea	86 (22.8)
Palpitations	74 (19.6)
Fatigue	72 (19)
Number of risk factors, median (IQR)	3 (2–4)
Smoking	148 (39.2)
Hypertension	282 (74.6)
Diabetes	130 (34.4)
Lipid Disorders	300 (79.4)
Obesity	158 (41.8)
Family history of coronary artery disease	152 (40.2)
Comorbidities	86 (22.8)
Peripheral angiopathy	22 (5.8)
Stroke	28 (7.4)
COPD	50 (13.2)
LVEF, mean (SD)	0.58 (0.05)
Coronary angiography	378 (100)
Left main coronary artery	0 (0)
Left anterior descending artery	128 (33.9)
Left circumflex artery	86 (22.8)
Right coronary artery	128 (33.9)
Angiographic score, median (IQR)	1 (0–2)
Therapy with cardioactive agents	272 (72)
Bruce protocol	154 (44)
Pharmacologic stress	196 (56)
Hard events	44 (11.6)
All-cause death	24 (6.3)
Cardiovascular death	14 (3.7)
Non-fatal myocardial infarction (post-scintigraphic study)	18 (4.8)
Soft events	120 (31.7)
Stroke (post-scintigraphic study)	20 (5.3)
Hospitalization due to heart disorder (post-scintigraphic study)	104 (27.5)
PTCA (post-scintigraphic study)	46 (12.2)
CABG (post-scintigraphic study)	8 (2.1)
Any cardiac event	138 (36.5)

CABG: coronary artery bypass graft surgery; PTCA: percutaneous transluminal coronary angioplasty.

**Table 2 medicina-59-01738-t002:** Descriptives of Summed Stress Score (SSS), Summed Rest Score (SRS), and Summed Difference Score (SDS).

Method	Index	Mean (SD)	Median (IQR)
ECTb	SSS	10.4 (5.9)	10 (5–14)
	SRS	4.9 (3.2)	4 (2–7)
	SDS	5.6 (3.9)	5 (2–8)
MYO	SSS	10.2 (5.6)	10 (5–14)
	SRS	5.4 (3.4)	5 (3–8)
	SDS	4.8 (3.4)	5 (2–7)
QPS	SSS	6.9 (3.9)	7 (4–9)
	SRS	3 (2.2)	2 (2–4)
	SDS	3.8 (3)	4 (1–6)
Expert scoring	SSS	5.4 (4)	4 (2–9)
	SRS	1.4 (0.9)	1 (1–2)
	SDS	4 (3.5)	3 (1–7)

ECTb: Emory Cardiac Toolbox; MYO: Myovation; QPS: Quantitative Perfusion SPECT.

**Table 3 medicina-59-01738-t003:** Roc analysis results for Summed Stress Score (SSS), Summed Rest Score (SRS), and Summed Difference Score (SDS).

Method	Index	AUC	95% CI	*p*	Optimal Cut-Off	Sensitivity (%)	Specificity (%)
ECTb	SSS	0.59	0.53–0.65	0.003	11.5	53.6	65.8
	SRS	0.54	0.48–0.6	0.241	-	-	-
	SDS	0.60	0.55–0.66	0.001	5.5	63.8	58.3
MYO	SSS	0.67	0.61–0.73	<0.001	10.5	68.1	63.3
	SRS	0.65	0.59–0.71	<0.001	4.5	69.6	53.3
	SDS	0.60	0.54–0.66	0.001	4.5	62.3	54.2
QPS	SSS	0.65	0.6–0.71	<0.001	6.5	66.7	59.2
	SRS	0.56	0.5–0.62	0.063	-	-	-
	SDS	0.66	0.6–0.71	<0.001	2.5	75.4	54.2
Expert scoring	SSS	0.88	0.84–0.91	<0.001	4.5	89.9	75.8
	SRS	0.72	0.67–0.77	<0.001	1.5	60.9	75.8
	SDS	0.87	0.83–0.91	<0.001	4.5	84.1	79.2

ECTb: Emory Cardiac Toolbox; MYO: Myovation; QPS: Quantitative Perfusion SPECT.

**Table 4 medicina-59-01738-t004:** Multiple Cox regression results for any cardiac event.

	Index	HR (95% CI) +	*p*
ECTb	SSS	1.03 (1.01–1.06)	**0.044**
	SRS	1.02 (0.97–1.07)	0.423
	SDS	1.06 (1.01–1.10)	**0.015**
MYO	SSS	1.06 (1.03–1.09)	**<0.001**
	SRS	1.11 (1.05–1.16)	**<0.001**
	SDS	1.06 (1.01–1.11)	**0.030**
QPS	SSS	1.08 (1.04–1.13)	**<0.001**
	SRS	1.07 (0.99–1.15)	0.084
	SDS	1.12 (1.06–1.19)	**<0.001**
Expert scoring	SSS	1.32 (1.27–1.38)	**<0.001**
	SRS	1.67 (1.42–1.97)	**<0.001**
	SDS	1.37 (1.31–1.44)	**<0.001**

ECTb: Emory Cardiac Toolbox; MYO: Myovation; QPS: Quantitative Perfusion SPECT; SDS: Summed Difference Score; SRS: Summed Rest Score; SSS: Summed Stress Score. Note. SSS, SRS, and SDS were entered in the model one at a time. + Hazard Ratio (95% Confidence Interval) adjusted for age, gender, number of risk factors, and comorbidity.

## Data Availability

Data available on request due to restrictions.

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
