# Peer review of "Long-Term Prognostic Value of Automated Measurements in Nuclear Cardiology: Comparisons with Expert Scoring"

_medicina, 2023, doi:10.3390/medicina59101738_

Round 1
Reviewer 1 Report
It is a good study. I think you can plan an image process trial about your topic.
Author Response
Dear Editor,
First, we would like to thank the reviewers for their kind and useful comments regarding our manuscript entitled ‘’LONG-TERM PROGNOSTIC VALUE OF AUTOMATED MEASUREMENTS IN NUCLEAR CARDIOLOGY: COMPARISONS WITH EXPERT SCORING’’, which was submitted for publication to your journal ‘’MEDICINA’’ (Kaunas).
All comments have been taken into account for the revision of the manuscript.
Comments – Reviewer 1
It is a good study. I think you can plan an image process trial about your topic.
Answer: Given the current research interest in artificial intelligence for medical imaging applications, we plan an image processing trial in the field of nuclear cardiology.
Yours Sincerely,
George Angelidis, MD-PhD
Reviewer 2 Report
I have some concerns for the aim of this study. AS far as I understand these software are supporting tools for the reporting.
From the discussion I understand that the use of these software is not fully validated. So which is the purpose to use them as a prognostic parameter.
Probably the manuscript is more suitable for a IT journal.
Author Response
Dear Editor,
First, we would like to thank the reviewers for their kind and useful comments regarding our manuscript entitled ‘’LONG-TERM PROGNOSTIC VALUE OF AUTOMATED MEASUREMENTS IN NUCLEAR CARDIOLOGY: COMPARISONS WITH EXPERT SCORING’’, which was submitted for publication to your journal ‘’MEDICINA’’ (Kaunas).
All comments have been taken into account for the revision of the manuscript.
Comments – Reviewer 2
I have some concerns for the aim of this study. As far as I understand these software are supporting tools for the reporting. From the discussion I understand that the use of these software is not fully validated. So which is the purpose to use them as a prognostic parameter. Probably the manuscript is more suitable for a IT journal.
Answer: In nuclear cardiology, the automated measurements of parameters, such as Summed Stress Score (SSS), Summed Rest Score (SRS), and Summed Difference Score (SDS, are world-widely taken into consideration in the evaluation of myocardial perfusion images in clinical practice. These automated measurements are obtained through the use of widely available corresponding software packages, and have an adjunctive role to visual evaluation of the scans.
In the manuscript under review, we presented our findings regarding the clinical role of the automated measurements of SSS, SRS, and SDS as long-term prognostic markers of morbidity and mortality, in comparison to the prognostic value of expert reading. These findings represent the last part of the research in our institution, concerning the clinical usefulness of automated analysis in patients who underwent myocardial perfusion imaging. Notably, our study was based on a long follow-up period (at least 36 months) and, to the best of our knowledge, provides unique evidence with regard to the prognostic value of automated measurements in nuclear cardiology.
On the other hand, software parameters and/or technical aspects of these software packages are out of the scope of our study. Therefore, we do believe that our manuscript is absolutely suitable to a clinical Journal (i.e. Medicina) and not to an IT journal.
Yours Sincerely,
George Angelidis, MD-PhD
Reviewer 3 Report
Dear editor;
I reviewed the article entitled ‘LONG-TERM PROGNOSTIC VALUE OF AUTOMATED MEASUREMENTS IN NUCLEAR CARDIOLOGY: COMPARISONS WITH EXPERT SCORING’. I found the article very interesting and useful for our journal. However, I have some minor comments before the acceptance of article.
--minor comments;
1-There are some grammatical mistakes in the article, hence, I recommend the proof-reading
2- The results section in the article is a lit bit long; therefore, it should be decreased.
3- Lastly, please cite this article: Value of C-reactive Protein/Albumin Ratio for Predicting Ischemia in Myocardial Perfusion Scintigraphy. Efe SÇ, Özdemir Candan Ö, GündoÄŸan C, Öz A, Yüksel Y, Ayca B, Çermik TF. Mol Imaging Radionucl Ther. 2020 Oct 19;29(3):112-117. doi: 10.4274/mirt.galenos.2020.88261.
4. Due to technical measurements, I think it would be beneficial to have them interpreted by a nuclear medicine doctor.
Dear author;
I reviewed the article entitled ‘LONG-TERM PROGNOSTIC VALUE OF AUTOMATED MEASUREMENTS IN NUCLEAR CARDIOLOGY: COMPARISONS WITH EXPERT SCORING’. I found the article very interesting and useful for our journal. However, I have some minor comments before the acceptance of article.
--minor comments;
1-There are some grammatical mistakes in the article, hence, I recommend the proof-reading
2- The results section in the article is a lit bit long; therefore, it should be decreased.
3- Lastly, please cite this article: Value of C-reactive Protein/Albumin Ratio for Predicting Ischemia in Myocardial Perfusion Scintigraphy. Efe SÇ, Özdemir Candan Ö, GündoÄŸan C, Öz A, Yüksel Y, Ayca B, Çermik TF. Mol Imaging Radionucl Ther. 2020 Oct 19;29(3):112-117. doi: 10.4274/mirt.galenos.2020.88261.
4. Due to technical measurements, I think it would be beneficial to have them interpreted by a nuclear medicine doctor.
Author Response
Dear Editor,
First, we would like to thank the reviewers for their kind and useful comments regarding our manuscript entitled ‘’LONG-TERM PROGNOSTIC VALUE OF AUTOMATED MEASUREMENTS IN NUCLEAR CARDIOLOGY: COMPARISONS WITH EXPERT SCORING’’, which was submitted for publication to your journal ‘’MEDICINA’’ (Kaunas).
All comments have been taken into account for the revision of the manuscript.
Comments – Reviewer 3
I reviewed the article entitled ‘LONG-TERM PROGNOSTIC VALUE OF AUTOMATED MEASUREMENTS IN NUCLEAR CARDIOLOGY: COMPARISONS WITH EXPERT SCORING’. I found the article very interesting and useful for our journal. However, I have some minor comments before the acceptance of article.
--minor comments;
1- There are some grammatical mistakes in the article; hence, I recommend the proof-reading
2- The results section in the article is a lit bit long; therefore, it should be decreased.
3- Lastly, please cite this article: Value of C-reactive Protein/Albumin Ratio for Predicting Ischemia in Myocardial Perfusion Scintigraphy. Efe SÇ, Özdemir Candan Ö, GündoÄŸan C, Öz A, Yüksel Y, Ayca B, Çermik TF. Mol Imaging Radionucl Ther. 2020 Oct 19;29(3):112-117. doi: 10.4274/mirt.galenos.2020.88261.
- Due to technical measurements, I think it would be beneficial to have them interpreted by a nuclear medicine doctor.
Answer:
1- We corrected the grammatical errors in the manuscript.
2- We have decreased the results section. Patients (demographic and clinical) characteristics are only presented in Table 1.
3- We included the above mentioned citation.
4- All studies (stress and rest images) were blindly evaluated by two independent experienced nuclear medicine doctors (expert scoring of SSS, SRS, and SDS). In parallel, measurements of SSS, SRS, and SDS were obtained by the three software packages, and recorded. Then, we compared expert scoring and the automated measurements of these parameters, as far as their prognostic significance.
Yours Sincerely,
George Angelidis, MD-PhD
Round 2
Reviewer 2 Report
Thank you very for the response, but my previous concerns regarding the aim and the clinical value of this study are still existed.